# Acute Toxic Effects of Tetrodotoxin in Mice via Intramuscular Injection and Oral Gavage

**DOI:** 10.3390/toxins15050334

**Published:** 2023-05-13

**Authors:** Fan Wang, Fuhai Zhang, Juxingsi Song, Shuaijun Zou, Jie Li, Yichao Huang, Liming Zhang, Qianqian Wang

**Affiliations:** Department of Marine Biomedicine and Polar Medicine, Naval Special Medical Center, Naval Medical University, Shanghai 200433, China; fanwang1130@126.com (F.W.); zhangfh1105@163.com (F.Z.); song9935@163.com (J.S.); sjzou@smmu.edu.cn (S.Z.); lijie1992@smmu.edu.cn (J.L.); ychuang@smmu.edu.cn (Y.H.)

**Keywords:** tetrodotoxin, acute toxic effects, muscle-strength testing, half lethal dose

## Abstract

Tetrodotoxin (TTX) is a highly fatal marine biotoxin. Constantly increasing intoxications and the lack of specific antitoxic drugs in clinical applications highlight the need for further research into the toxic effects of TTX. Current reports on poisoning cases and the TTX toxicity mechanism suggest that the blocking of voltage-gated sodium channels (VGSCs) by TTX is probably reversible, but direct evidence of this is lacking, as far as we are aware. This study explored the acute toxic effects of TTX at sub-lethal doses via different routes, analyzing variations in muscle strength and TTX concentration in the blood in mice. We found that the loss of muscle strength in mice caused by TTX was dose-dependent and reversible, and the death time and muscle strength variations after oral gavage with TTX appeared to occur later and were more variable than those after intramuscular injection. In conclusion, we systematically compared the acute toxic effects of TTX for two different administration routes at sub-lethal doses, directly verifying the reversible reaction of TTX blocking VGSCs and speculating that averting a complete block of VGSCs by TTX could be an effective strategy for preventing death from TTX poisoning. This work may provide data for the diagnosis and treatment of TTX poisoning.

## 1. Introduction

Tetrodotoxin (TTX) is a potent marine biotoxin that blocks voltage-gated sodium channels (VGSCs) and reduces neuroexcitability [1]. TTX can cause limb paralysis, paresthesia and even cardiorespiratory failure in humans [1,2]. In cases of severe TTX poisoning, death often occurs within hours; however, only symptom-based supportive care is available due to a lack of effective antidotes [3].

TTX is produced by bacteria such as *Pseudomonas* and *Vibrio* spp. [4]. It was first found in puffer fish, as it can be enriched in animals through the food chain [4,5]. TTX poisoning in humans used to be associated with puffer fish consumption, and most intoxications were limited to East Asia because of the food culture there [6]. In recent years, however, TTX has been found in other species, such as bivalves and gastropods, which are important sources of animal protein food in coastal areas [7,8]. Its geographical distribution has expanded to various regions of the Mediterranean Sea, and the Pacific and Indian Oceans [1]. Existing regulations worldwide only restrict trading of the fish that generally contain TTX, such as puffer fish, yet the monitoring of TTX in other sea food has not been normalized, resulting in numerous TTX poisoning incidents [7]. TTX is jeopardizing food safety and has become a nonnegligible problem.

On the other hand, TTX is also becoming an important tool in VGSC research owing to its high selectivity and strong activity toward VGSCs: it has great potential for use in local anesthesia, analgesia and treatment of arrhythmia [9,10,11]. TTX obtained through artificial synthesis and extraction has entered the commercial market legally [12,13]; however, there is an additional—and fatal—risk that TTX could be used in illegal ways. Intramuscular injection and oral poisoning could be possible routes for illegal use of TTX; however, corresponding toxicity data are rare [1,14,15].

We, therefore, decided to explore the acute toxic effects of TTX within the range of sub-lethal doses and focused on the routes of intramuscular (i.m.) injection and oral gavage (o.g.) in order to provide further reference for diagnosis and treatment of TTX poisoning.

## 2. Results

### 2.1. TTX Poisoning Models

TTX poisoning models were established by i.m. injection and o.g. administration in ICR (Institute of Cancer Research) mice. Toxic symptoms and muscle strength changes in the mice were recorded. It was found that the muscle strength in TTX poisoning by i.m. injection declined obviously with the increase in TTX dosages until death occurred (Figure 1A). At the 30 min time point after TTX administration, 4.5 μg/kg of toxin resulted in a reduction of 59.7 g in muscle strength compared with the control group, and the muscle strength in the 7.5 μg/kg group decreased by 147.0 g. In the 9.0 μg/kg group, 4/6 mice died 12~15 min after TTX administration, and the muscle strength of the surviving mice decreased by 111.8 g. 

For TTX poisoning by o.g. administration, a weaker correlation between muscle strength decrease and TTX dose increase was observed (Figure 1B). Considering that residual food in the gastrointestinal tract may affect the absorption of TTX, the daily ration of mice was evaluated thoroughly (Appendix A). Based on the assessment results, food intake was maintained at 3 g per day in the o.g. model to reduce the effect of residual food on TTX absorption.

### 2.2. Acute Toxicity of TTX

In order to clarify the acute toxicity of TTX, the half lethal doses (LD_50_) of TTX poisoning by both the i.m. and o.g. routes were determined using a modified up-and-down procedure in the models. Figure 2 presents all the results in a semi-logarithmic scale as the percentage of mice mortality versus toxin doses. The estimated LD_50_ values of TTX were 8.6 μg/kg by i.m. injection and 378.7 μg/kg by o.g. administration, implying that toxicity by o.g. administration was approximately 44 times lower than that by i.m. injection.

During the acute toxicity test, all deaths occurred within 2 h; other mice survived to 48 h (Table 1). However, at the absolute lethal doses, the survival time for TTX i.m. injection (10 μg/kg) was only 12~20 min, while the survival time for TTX o.g. administration (600 μg/kg) was 14~70 min (Figure 3). Seizures were only observed before death (Table 1). All poisoned mice developed apathy and various degrees of paralysis; the latter was apparently affected by TTX dose, but it could not be well graded through manual muscle testing [16].

### 2.3. TTX Concentration in Blood following i.m. and o.g. Administration

To explore the toxicokinetics of TTX, free TTX in the blood of poisoned mice was determined using the ultra-performance liquid chromatography–tandem mass spectrometry (UPLC-MS/MS) method (Appendix A); the results are shown in Figure 4. Doses of 7 µg/kg by i.m. and 300 µg/kg by o.g. were chosen, corresponding to 0.8× LD_50 (i_._m_._)_ and 0.8× LD_50 (o_._g_._)_, because these doses of TTX cause fewer deaths in mice, but the TTX concentration is still detectable in the blood. In the poisoning model using i.m. injection, the concentration of TTX in the blood was 11.56 ng/mL at 5 min after TTX administration, after which it declined gradually. After 1 h, TTX could barely be detected in the blood. However, in the o.g. administration model, the TTX concentration gradually increased to 9.41 ng/mL at 45 min, then declined to 4.44 ng/mL at 2 h. Compared with i.m. injection poisoning, there was a delayed absorption and elimination of TTX after o.g. administration.

### 2.4. Muscle Strength Changes after TTX Poisoning

To accurately reflect the dynamic changes in paralysis after TTX poisoning, muscle strength changes in the mice were measured under a sequence of sub-lethal doses (0.9× LD_50_, 0.7 LD_50_ and 0.5× LD_50_). The mice injected with sub-lethal doses of TTX showed a sharp reduction in muscle strength within 30 min, then gradually recovered within 24 h (Figure 5A–D). At the 30 min time point, the muscle strength of the mice exposed to 0.9× LD_50_ of TTX fell from 202.1 g to 64.5 g, while the muscle strength of the mice in the 0.7× LD_50_ and 0.5× LD_50_ groups was reduced to 130.6 g and 151.8 g, respectively. The mice injected with 0.5× LD_50_ of TTX showed a faster recovery and their muscle strength was restored to normal within 3 h. Altogether, the degree of muscle strength decline was significantly related to the TTX exposure dose.

The mice receiving sub-lethal doses of TTX by o.g. administration exhibited falling–rising oscillations in muscle strength (Figure 5E–H). In the 0.9× LD_50_ group, muscle strength dropped from 236.9 g to 133.9 g at the 1 h time point, while in the 0.7× LD_50_ and 0.5× LD_50_ groups, muscle strength loss was not significant throughout the process. Three of nine mice died within 2 h when dosed with 0.9× LD_50_ or 0.7× LD_50_ of TTX via o.g. administration. Dead mice suffered more muscle strength loss than surviving mice at the 1 h time point (Appendix A). In general, muscle strength loss in TTX poisoning via o.g. administration was also reversible and dose-dependent, but showed a delayed peak, compared with that via i.m. intoxication.

### 2.5. Pathological Examination

Tissue damage after TTX poisoning was evaluated via pathological examination. Compared to the normal group, the hearts, livers, spleens, lungs, kidneys, stomachs, ileums, colons and brains of the mice that were administered with 0.9× LD_50_ of TTX by both administration routes did not manifest obvious pathological changes (Figure 6 and Figure 7).

## 3. Discussion

TTX is a highly fatal marine biotoxin. In recent years, its growing threat to food safety has attracted much attention [8], and the possibility of its illegal application has also been noted [14]. Consequently, the increasing intoxications and lack of specific antitoxic drugs in clinical applications highlight the need for further research into the toxic effects of TTX. This study explored the acute toxic effects of TTX in mice at sub-lethal doses by analyzing the variations in muscle strength and TTX concentrations in blood via different routes. The dynamic changes in TTX toxic effects were emphasized, in contrast to previous studies [1]. Based on the variations in muscle strength, poisoning symptoms and death time of TTX-poisoned mice, our work could provide data for the diagnosis and treatment of TTX poisoning.

Tetrodotoxin is known as a robust neurotoxin. It can block the inflow of sodium ions by binding to VGSCs, impeding the entry of Na^+^ ions, which are crucial for resting membrane potential and for neuronal excitability [17,18]. VGSCs have nine different isoforms (Na_v_1.1~1.9) in the mammalian nervous system [19]. Of these isoforms, six (Na_v_1.1, Na_v_1.2, Na_v_1.3, Na_v_1.4, Na_v_1.6 and Na_v_1.7) have been identified as sensitive to TTX, with an IC_50_ approximately equal to 10 nM. These six isoforms are predominantly expressed in skeletal muscles and the nervous system, and are not involved in excitation phenomena in healthy adult hearts. In contrast, Na_v_1.5, Na_v_1.8 and Na_v_1.9 control the electrophysiological activities of the heart and the dorsal root ganglion with an IC_50_ above or equal to 1 µM. As a low dosage of TTX can barely cross the blood–brain barrier [20,21], in a certain dose range, TTX predominantly affects the conduction of the peripheral nervous system and skeletal muscles via the blocking of Na_v_1.4, Na_v_1.6 and Na_v_1.7 [19]; thus, the influence of TTX on the peripheral nervous system and skeletal muscles requires further investigation. Previous studies showed a trend for self-healing in TTX poisoning [22,23,24,25], but the process lacked detailed descriptions. Therefore, we established TTX poisoning models via i.m. injection and o.g. administration in mice to examine the dynamic process of TTX intoxication.

First, the LD_50_ values of TTX in mice were estimated via different routes. The estimated LD_50 (o_._g_._)_ in the present study was close to those found by Abal [26] and Xu [15]; the minor differences may have been caused by toxin samples from different sources or different experimental designs. Unfortunately, the LD_50 (i_._m_._)_ in mice has not yet been found elsewhere, as far as we know, but ours was close to the LD_50_ value of subcutaneous and intraperitoneal injections [15]. Symptoms such as limb paralysis, apathy and seizures were observed in poisoned mice, and limb paralysis was found to be the dominant manifestation of TTX poisoning. Different doses of TTX were further selected to quantitatively analyze the changes in muscle strength in mice after poisoning. It was found that the loss of muscle strength in mice caused by TTX was dose-dependent and reversible, which verified that the blocking of VGSCs by TTX is reversible. Massive muscle-strength loss in mice before death indicated that respiratory muscle inhibition could be one of the main causes of the death of mice from TTX poisoning.

Since 1941, the estimation of TTX poisoning has been guided by a clinical grading system largely based on symptoms [1]. However, the grading system is greatly affected by subjective judgment and the proficiency of clinicians, and it cannot adequately describe the dynamics of TTX poisoning. In this study, we found that changes in muscle strength can reflect the trends of TTX poisoning. As far as we are aware, this is the first time the reversible change process of symptoms in TTX poisoning has been quantified. The measuring of muscle strength change has great potential to be a reference indicator in toxicity study, antidote evaluation and clinical diagnosis of TTX poisoning.

Both absorption and excretion of TTX are rapid in vivo [21]. In order to establish the relationship between toxin absorption and muscle strength decline, the concentrations of TTX in blood after poisoning were measured. We found that muscle strength significantly declined approximately 15~30 min after the TTX blood concentration reached its peak, and then gradually recovered along with the metabolism of TTX. This indicates that, in vivo, the reduction in muscle strength is correlative to the concentration of TTX. Thus, we speculate that TTX blocks the VGSCs of the respiratory muscles and their afferent motor neurons at a low dosage, inhibiting the contraction of the respiratory muscles, which then leads to cerebral hypoxia and death in mice. Seizures before death may result from cerebral hypoxia.

TTX is mainly distributed in the heart, lungs, stomach and kidneys after poisoning, and is excreted through the kidneys [21]. Other work by Abal suggests that oral TTX causes ultrastructural cell damage to the liver, spleen and intestines at 2 h [27]. In our work, however, pathological examination indicated that acute TTX poisoning at sub-lethal doses did not cause obvious damage to primary organs in mice at 24 h. Taken together, we speculate that the gastrointestinal effects induced by TTX may be transient and might not develop further. Although the LD_50_ of o.g. administration was approximately 44 times higher than that of i.m. injection, there was no significant difference in peak TTX plasma concentration between the two routes, suggesting a low bioavailability of TTX. Considering the rapid metabolism of TTX in mice, the steep slope of the dose–mortality curve of TTX via i.m. injection (Figure 2A) implied that patients could survive TTX poisoning as long as the VGSCs are not fully blocked. Based on these results, we hypothesize that if the absorption of TTX is reduced or delayed, then the blocking of TTX on VGSCs is maintained at a non-lethal level, and in that way some deaths may be avoided. Thus, current first-aid measures in the early stage, such as gastric lavage and adequate basic life support, are essential in the treatment of TTX poisoning.

Death and the reduction in muscle strength caused by the o.g. administration of TTX occurred later than these outcomes via i.m. injection, suggesting that the absorption of TTX in the gastrointestinal tract might delay the toxic effect of TTX, even in starvation. The survival time for oral TTX poisoning at the absolute lethal dose level in mice varied from minutes to hours, hinting at a clinically relevant time window for emergency treatment. In contrast, TTX poisoning via i.m. injection seems to be more difficult to rescue due to its fast onset. Considering the acute onset and high mortality of TTX poisoning, strict controls for the preparation and marketing of TTX are urgently needed, and effective antagonists must be developed.

## 4. Conclusions

In conclusion, we compared the acute toxic effects of TTX in mice for i.m. injection and o.g. administration; we also verified, through muscle-strength testing and the monitoring of TTX concentrations in blood, that the blocking of VGSCs by TTX is dose-dependent and reversible. The inhibition of respiratory muscles may be the main cause of death in lower-dose TTX poisoning. Averting a complete block of VGSCs by TTX could be an effective strategy to prevent death in cases of TTX poisoning. These results provide new insights into the acute toxic effects of TTX.

## 5. Materials and Methods

### 5.1. Reagents

Tetrodotoxin (HPLC 99%) was supplied by Zhongyang Biotechnology Co., Ltd. (Shanghai, China). Formic acid, acetic acid and acetonitrile from Sinopharm Group Chemical Reagent Co., Ltd. (Shanghai, China) were analytical-grade. Phosphate buffer saline (PBS) and paraformaldehyde fixative were purchased from Servicebio Technology Co., Ltd. (Wuhan, China). Glucose serum (5%) was from Chenxin Pharmaceutical Co., Ltd. (Jining, China). Primary secondary amine (PSA) and capped octadecyl silica gel (C18) powder were obtained from Agela Technologies (Torrance, USA). Distilled water was purified using a water purification system (Elix, Merck Millipore, Molsheim, France). Food for mice in conformity with Chinese national standard (GB 14924.3-2010) was purchased from the Laboratory Animal Center at the Naval Medical University (Shanghai, China).

### 5.2. Animals and Poisoning Model

Experiments were performed using adult Institute of Cancer Research (ICR) mice (18~22 g) purchased from the Laboratory Animal Center at the Naval Medical University (Shanghai, China). All procedures were performed according to the Guide for the Care and Use of Laboratory Animals. Mice were acclimatized to the laboratory for at least 3 days before experiments started. Mice were housed with free access to water under controlled temperatures (20~24 °C) and relative humidities (45~65%).

TTX was initially dissolved in 0.1% (*v*/*v*) aqueous acetic acid and was diluted in PBS solution. Working concentrations were set as 1 μg/mL (i.m.) and 50 μg/mL (o.g.). Mice were given a single dose of TTX by o.g. or i.m. injection to the right leg. For the TTX o.g. model, 3 g food per mouse was given every morning at 8:00 a.m., and the mice were fasted 12 h prior to the o.g. administration of TTX and 3 h after the administration. Glucose serum (5%) was administered ad libitum during the 12 h fasting period before the TTX o.g. administration. The mice in the i.m. model had free access to food.

### 5.3. Experimental Design

#### 5.3.1. Determination of the Lethal Dose 50 (LD_50_)

A modified up-and-down procedure was established to determine the LD_50_ [26,28]. Doses were selected from the sequence 17, 13, 10, 9, 8, 7, 6, 5 μg/kg for i.m. injection or 1000, 600, 500, 400, 325, 250, 150, 100 μg/kg for o.g. administration. The first group of 10 mice (5 males and 5 females) was dosed at 8 μg/kg and 600 μg/kg, respectively. Symptoms were observed for a total of 48 h. Time of death was indicated by last gasping breath. If the death rate was above 50%, the next group would receive a lower dose, and vice versa. The number of mice was equal at each level. At least 5 levels of doses were tested to estimate the LD_50_. Calculations were performed using GraphPad Prism (Version 9.3.1, GraphPad Software, Inc, San Diego, CA, USA).

#### 5.3.2. Blood Sample Collection and Measurement of Blood TTX

For the toxicokinetic studies of TTX i.m. injection, groups of mice (3 males and 3 females in each group) were administered with TTX 7 µg/kg, while for TTX o.g. administration, each mouse received a dose of 300 µg/kg. Serial serum samples (0.5 mL) were obtained via eyeball blood collection 0.083, 0.167, 0.333, 0.5 and 1 h after i.m. injection and 0.25, 0.5, 0.75, 1, 1.5 and 2 h after o.g. administration, and then stored frozen at −80 °C until analysis.

Serum (100 µL) was mixed with 2% (*v*/*v*) aqueous acetic acid to achieve 1 mL, and vortexed for 3 min. After a water bath (100 °C) for 5 min, 50 mg PSA and 50 mg C18 were added and fully mixed. The mixture was then bathed in ice for 5 min and centrifuged at 2655× *g* for 10 min. The supernatant passed through a 0.2 µm filter and was transferred into a sample vial for UPLC-MS/MS analysis.

Analyte separations were performed on a Nexera LC-40B XR system (Shimadzu Corp., Tokyo, Japan) using a Diol-HILIC-120 column (2.1 × 100 mm, 1.9 µm, Shimadzu Corp., Tokyo, Japan) equilibrated with 2% solvent A (aqueous formic acid, 0.1% *v*/*v*) and 98% solvent B (acetonitrile) at a flow rate of 0.42 mL/min. After 0.5 min, solvent A was gradually increased to 90% over 3.5 min and maintained at 90% for 1.0 min. Solvent A was then set back to 2% immediately. The column temperature was set at 40 °C, and the injection volume was 10 µL.

Analyses of TTX in serum samples were performed using a LCMS-8045 system (Shimadzu Corp., Tokyo, Japan) equipped with electrospray ionization (ESI). Quantitative analysis of TTX was performed in the positive ion mode under the following conditions: ion spray voltage at 4000 V, atomizer flow at 3 L/min, dryer flow at 10 L/min, heater flow at 10 L/min, desolvation temperature at 526 °C and scan type MRM (multiple reaction monitoring). The ion transitions used were 320.2 > 301.8/162.0 (*m*/*z*), with collision energy at 27 and 41 eV, respectively.

#### 5.3.3. Muscle-Strength Testing

Muscle-strength testing was performed using a grip strength meter (XR-YLS-13A, Lowe Biotechnology, Shanghai, China). Each mouse was placed on the grip board individually and was pulled back gently by the tail. When the mouse held the bars tightly, it was pulled back with constantly increasing force, until its paws were loosened. The device would automatically record the animal’s maximum gripping force. Each mouse was tested 3 times and the average value was taken.

#### 5.3.4. Experimental Conditions

Male ICR mice (6 per group) were used to estimate the acute toxic effects of TTX. Three levels of doses (0.9× LD_50_, 0.7× LD_50_ and 0.5× LD_50_) were administered via both routes. Muscle-strength testing was conducted at 0.25 h before and 0.5, 1, 2, 3, 6, 12 and 24 h after TTX administration. Three extra mice were dosed with TTX via o.g. in the 0.9× LD_50_ and 0.7× LD_50_ groups, due to deaths occurring during the muscle strength test. Mice that survived this experiment were euthanized via CO_2_ inhalation. Samples of several organs (heart, liver, spleen, lung, kidneys, stomach, ileum, colon and brain) were collected and subjected to pathological examination.

### 5.4. Statistical Analysis

Statistical analysis was performed using IBM SPSS Statistics (Version 26.0.0.0, International Business Machines Corp., Armonk, USA) and GraphPad Prism. The data for muscle strength are presented as the mean ± standard deviation (SD). Survival time is presented as plots and median. The dispersion of TTX blood concentration was visualized in a box plot, and the whiskers extended to the minimum and maximum values. Analysis of variance was used for the evaluation of the repeated measurement data and further pairwise comparison was conducted using the Bonferroni test. Statistical significance in survival time was evaluated using the Mann–Whitney test.

## Figures and Tables

**Figure 1 toxins-15-00334-f001:**
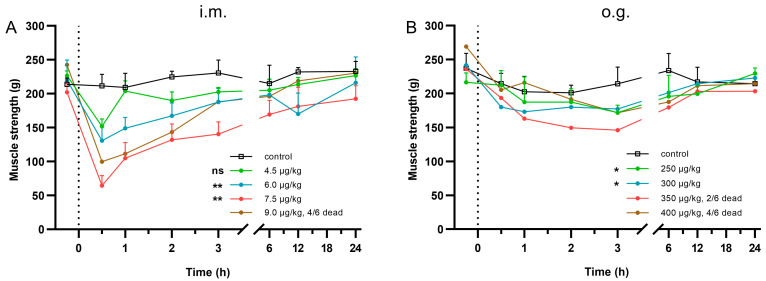
Muscle strength changes after tetrodotoxin (TTX) poisoning in mice (*n* = 6). TTX was administered via intramuscular (i.m.) injection (**A**) or oral gavage (o.g.) (**B**) at time 0. Curves indicate the gripping force of the surviving mice at each time point. Data are expressed as mean ± standard deviation (SD). ns, no significance, * *p* < 0.05, ** *p* < 0.01 compared with control group.

**Figure 2 toxins-15-00334-f002:**
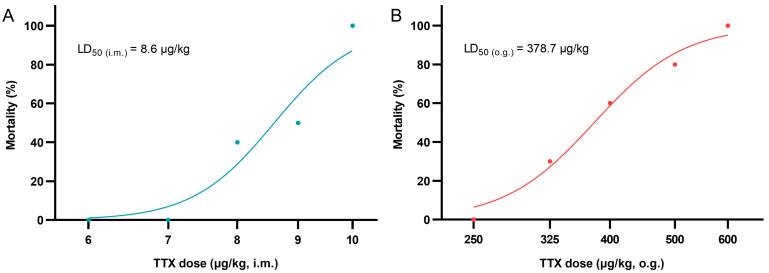
Dose–response curves for TTX-induced mortality by i.m. (**A**) and o.g. (**B**) routes in mice (*n* = 10). Percentage lethality values were plotted against the concentrations of TTX. The LD_50_ value was estimated using a nonlinear regression fitting procedure. R^2^ were 0.9213 and 0.9830, respectively.

**Figure 3 toxins-15-00334-f003:**
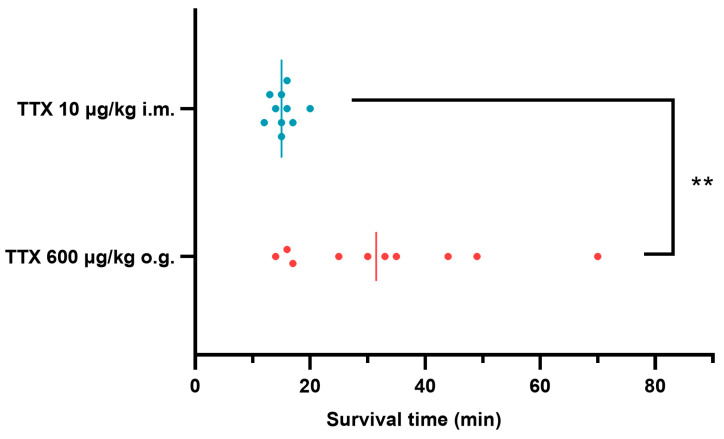
Survival time comparison between i.m. and o.g. administration of TTX at respective absolute lethal dose level in mice (*n* = 10). The median survival times represented by the two vertical lines in the graph are 15.0 min and 31.5 min, respectively. ** *p* < 0.01 comparison between the two groups.

**Figure 4 toxins-15-00334-f004:**
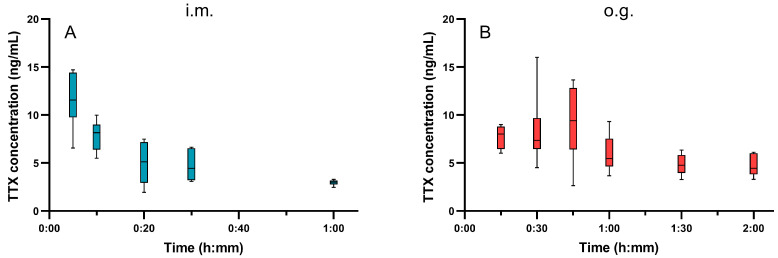
Time course of TTX concentration in the blood of mice after TTX i.m. injection (**A**) or o.g. administration (**B**) (*n* = 6).

**Figure 5 toxins-15-00334-f005:**
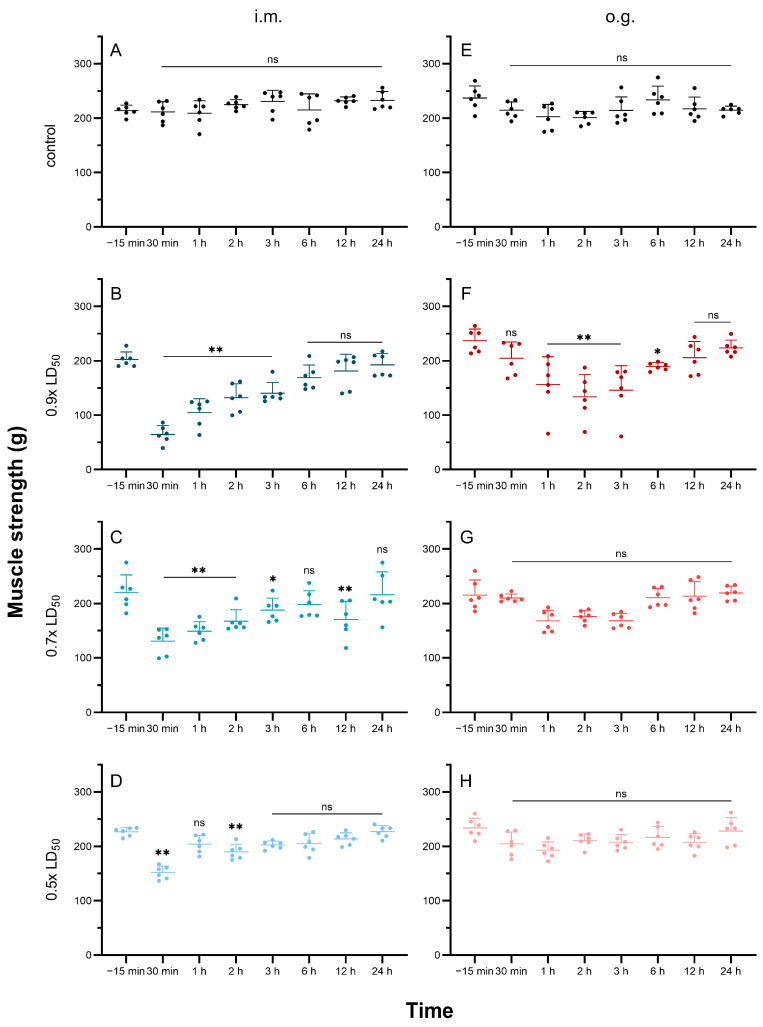
Muscle strength loss after TTX intoxication in surviving mice (*n* = 6). Different doses of TTX were administered via i.m. (**A**–**D**) or o.g. (**E**–**H**) routes. Data are expressed as mean ± SD. ns, no significance, * *p* < 0.05, ** *p* < 0.01 compared with levels before TTX administration (time—−15 min).

**Figure 6 toxins-15-00334-f006:**
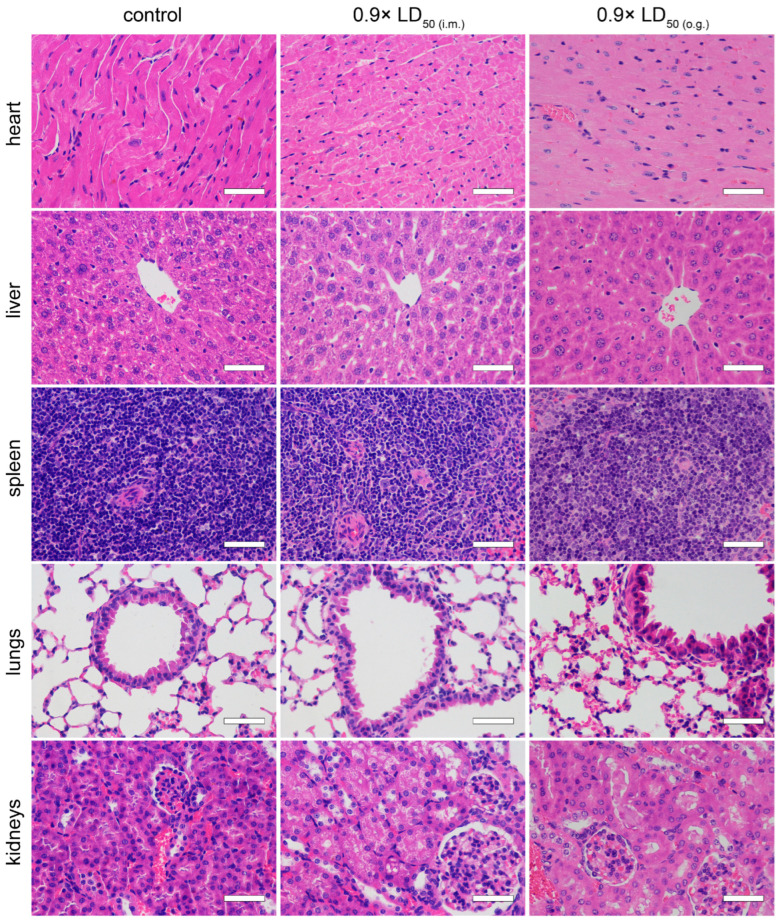
Pathological changes in organs (heart, liver, spleen, lungs and kidneys) of mice 24 h after TTX poisoning (*n* = 3). Scale bar = 50 μm.

**Figure 7 toxins-15-00334-f007:**
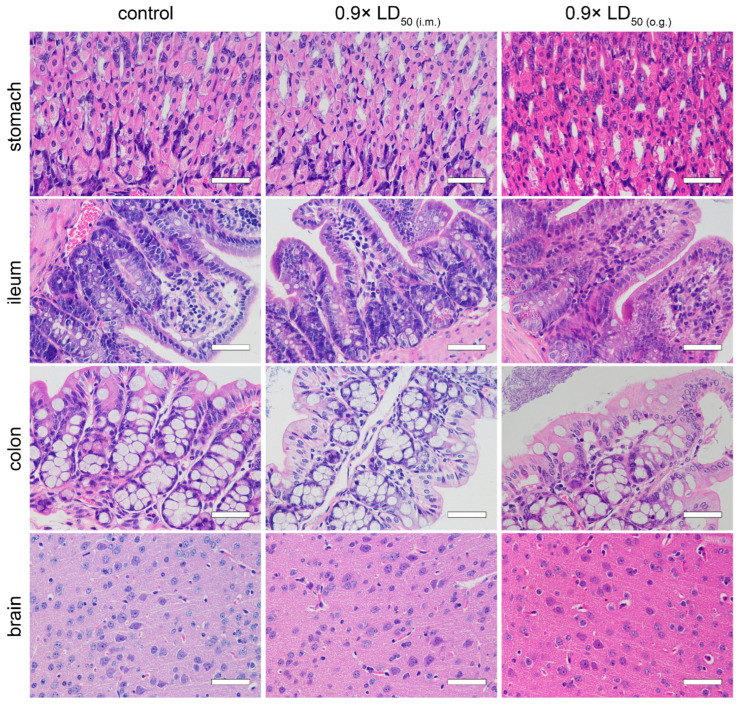
Pathological changes in organs (stomach, ileum, colon and brain) of mice 24 h after TTX poisoning (*n* = 3). Scale bar = 50 μm.

**Table 1 toxins-15-00334-t001:** Toxic symptoms and mortality in mice after TTX poisoning via different administration routes (*n* = 10).

Dose (μg/kg)	Toxic Symptoms		Mortality
	Apathy	Seizures	
Intramuscular injection			
10	10/10	10/10	10/10
9	10/10	5/10	5/10
8	10/10	4/10	4/10
7	10/10	0/10	0/10
6	10/10	0/10	0/10
Oral gavage			
600	10/10	10/10	10/10
500	10/10	8/10	8/10
400	10/10	6/10	6/10
325	10/10	3/10	3/10
250	10/10	0/10	0/10

## Data Availability

The data presented in this study are available in this article and Appendix A.

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
