# Peer review of "Acute Toxic Effects of Tetrodotoxin in Mice via Intramuscular Injection and Oral Gavage"

_toxins, 2023, doi:10.3390/toxins15050334_

Round 1

Reviewer 1 Report

Manuscript entitled „ cute Toxic Effects of Tetrodotoxin at Sub-lethal Doses: A Comparison between Intramuscular Injection and Intragastrical Administration” is very interesting, well-written and well-planned experimental work. I fully support the publication of this manuscript; however I recommend some small corrections according the comments:

Introduction

Results

Line 55 – explain in full name an abbreviation ICR

Figure 1 – explain for the first-time abbreviations TTX, i.m. and i.m.

use to mark the different colours on the graph corresponding to the appropriate TTX dose. At the moment, the use of different shades of one colour is hardly legible.

Line – 91 – “In this study, all deaths occurred within 2 h, and other mice survived to 48 h”. explain exactly in the text of the manuscript what the mortality was at the specific doses of TTX used (number of animals versus time of death after administration of a certain dose). Unfortunately, the graph does not show this clearly and understandably.

Materials and Methods

-        complete information on the number of animals used in individual experiments where it is not described

Reviewer 2 Report

The article “Acute Toxic Effects of Tetrodotoxin at Sub-lethal Doses : A Comparison between Intramuscular Injection and Intragastrical Administration” sought to evaluate the differences in acute toxicity of TTX on mice depending on the route of administration. Not surprisingly, the dietary route (intragastric exposure) is much less toxic than the intramuscular route. The authors describe the symptoms observed, the toxicokinetics, the effects of TTX on muscle contraction.

Their results are fairly predictable, and the study adds little new knowledge about the acute toxicity of TTX. 

The article should be restructured to present acute toxicity with LD50s and symptoms (Figure 3 + Table 1), toxicokinetics (Figure 8), then effects on muscle contraction (Figure 1 and Figure 5) and finally Figures 6 and 7 (which are not very useful). Figures 2 and 4 are useless.

Major

Figure 1: the colors of the curves must be changed because it is difficult to distinguish the curves from each other on the two graphs. Use very different colors on the same graph (red, blue, green, grey, brown). Statistics: on the graphs, you cannot see where the significant differences appear.

Figure 2 should simply be a supplementary figure. It does not add anything to the article because it does not involve TTX.  L71-79 should not be placed in the heart of the article: it is simply a methodological aspect.

Figure 3: LD50im = 8.593 µg/kg and LD50ig = 378.7 µg/kg. Harmonize the number of digits after the decimal point (1 or 3).

Figure 4 is useless: the survival time can be included in Table 1.

Figure 3 and Table 1 should appear at the beginning of the results. It would make more sense to have the acute toxicity with symptoms and LD50 values, then the muscle strength study afterwards. So Figure 3 should become Figure 1.

Similarly figure 8, which gives an indication of TTX toxicokinetics should appear as figure 2.

In the discussion, the authors should compare their LD50ig values with those published elsewhere, especially in reference 27: the values are 378.7 in this article and 232 µg/kg in ref. 27. Similarly, you should compare the LD50im published here (8.6 µg/kg) with published values. This point is very important to consider because your results must be discussed.

Minor

The title of the article could be shortened to "Acute Effects of Tetrodotoxin using Intramuscular and Intragastrical Administration" especially since the doses used are not systematically sublethal.

L11 : via different routes

L21 : This study provides

L28 : Death often occurres

L31-32 : TTX is produced by bacteria such as Pseudomonas and Vibrio spp. It was first found in puffer fish, as it can be enriched in animals through the food chain.

L39: yet the monitoring of TTX

L39: in other sea food has

L57: TTX poisoning by i.m. injection

L169: the entry of Na+ ions

L174: expressed in skeletal muscles and the nervous system

L185: the TTX LD50

L185: by different routes

L186: The symptoms

The English language expression is quite poor and could be significantly improved if a proofreader fluent in English reviewed the article. This is important before publication.

Reviewer 3 Report

General and Detailed Comments

This manuscript adds very little to what is already known on the acute toxic effects of tetrodotoxin, and to previous work that has been performed with the neurotoxin. In addition, there are a number of points that merit to be clarified and revised, and which make the manuscript unsuitable for publication in its present form. Starting from the title (lines 2-3):

1.       Lines 2-3, The title is misleading, because according to Material and Methods section, TTX was administered by oral gavage and not by intragastrical administration. What is the proof of this assertion? Intragastrical administration means administering directly into the stomach of mice. In any case the intragastric delivery system should be described, if used, otherwise, the title must be changed.

2.       Line 7, add "into ""the"" toxic

3.       Line 13, the term  «  intragastrical » is not correct (see comment 1)

4.       Line 244, LC-MS chromatogram of TTX standard, and the Tetrodotoxin (HPLC 99%) used should be presented

5..        Line 260 it should be  " disolved"  in 0.1%

6.       Line 263, it should be specified what kind of gavage was used "oral” gavage"

7.       Line 264, it is unethical, unfair, and wrong to fasten a mouse weighing 18-22 g for "12 h" , since there is evidence that in addition to the loss of weight superior to 10%, a number of blood parameters are changed. 

8.       Line 265, it should be defined what is meant by intra gastric administration (i.g.) and how is was done. In addition, it is not clear how the food intake was limited to 3 g.

9.       The way how muscle strength changes after TTX poisoning is not the better way to assess the skeletal muscle capacity

10.   The Discussion is quite limited and the Results poorly discussed, and no interpretation or answers are given for results presented. No discussion with the paper previously publiched on the subject? see:

Abal P, Louzao MC, Vilariño N, Vieytes MR, Botana LM. Acute Toxicity Assessment: Macroscopic and Ultrastructural Effects in Mice Treated with Oral Tetrodotoxin. Toxins (Basel). 2019. 11(6):305. doi: 10.3390/toxins11060305. PMID: 31146400; PMCID: PMC6628385.

11.   Line 162, “lack of effective antidotes”. Fortunately, a number of monoclonal and polyclonal can be used as antidotes.

12.   Lines 166-167, The assertion “Our work may provide references for the diagnosis, treatment and prevention of TTX poison” based on the present manuscript is misleading.

The quality of English language of the manuscript is OK, but can be improved. 

Round 2

Reviewer 2 Report

The authors have taken into account all the remarks that were made to them. Many changes have been made to the draft. The article is restructured and much easier to read.

The English language expression has been improved. There are still some errors that can be changed during the editing of the article.

Reviewer 3 Report

The manuscript has been considerably improved. Furthermore, the Animal Ethical issues raised previously have been explained by additions in the text. The supplementary results are adequate. Consequently, the manuscript is now suitable for publication, even if some points merit revision as indicated here below:

   Lines 1-2 Please add animal species in both title (Acute mouse Toxic Effects….) and in the key contribution

2.      Lines 260-261, the changes performed have no sense.

I suggest “Tetrodotoxin is known as a robust neurotoxin. It can block the inflow of sodium ions by binding to VGSC impeding the entry of Na+ ions, which is crucial to the resting membrane potential and for neuronal excitability”

3.      Lines 388-390, the term “demonstrates that patients” is misleading

Considering the rapid" mouse" metabolism of TTX in vivo, the steep slope of the dose–mortality curve of TTX by i.m. injection (Figure 2A) “demonstrates that patients” could survive from TTX poisoning as long as VGSCs are not fully blocked.

“demonstrates that patients” is inappropriate, you cannot extend what you have found in mouse experiments to humans, at most you can suggest. Please amend sentence.
